# Introducing Molecular Sieve into Activated Carbon to Achieve High-Effective Adsorption for Ethylene Oxide

**DOI:** 10.3390/nano14181482

**Published:** 2024-09-12

**Authors:** Feng Liu, Lingyan Qin, Pingwei Ye, Bo Yang, Qiong Wu, Li Li, Yuwei Dai, Chuan Zhou, Sumin Li

**Affiliations:** 1School of Materials Science & Engineering, Jiangsu University, Zhenjiang 212013, China; liufeng001013@163.com (F.L.); dywei510@163.com (Y.D.); 2State Key Laboratory of NBC Protection for Civilian, Beijing 102205, China; qinlyjy@163.com (L.Q.); dahema2007goodluck@163.com (B.Y.); wuqiong1710@aliyun.com (Q.W.); lily97@buaa.edu.cn (L.L.); 3College of Chemical Engineering, Beijing University of Chemical Technology, Beijing 100029, China

**Keywords:** activated carbon, ZSM-5, ethylene oxide, adsorption, mechanism

## Abstract

Presently, ethylene oxide (EtO) is posing a significant threat to both human health and the environment due to occasional or deliberate emissions. However, few works so far have focused on this issue. It is urgent to explore novel and effective technology to protect against the threat of EtO. Herein, a series of AC/ZSM-5 composites were prepared to improve the adsorption performance for EtO, evaluated by dynamic breakthrough experiments. Particularly, the AC/ZSM-20% composite demonstrated a more excellent adsorption capacity of 81.9 mg/g at 25 °C and 50% RH than that of pristine AC and ZSM-5 with 32.5 and 52.3 mg/g, respectively. Moreover, the adsorption capacity of the AC/ZSM-20% composite remained constant even after five adsorption-desorption cycles. The adsorption mechanism of EtO on the composite is further revealed by density functional theory (DFT) calculations.

## 1. Introduction

Ethylene oxide (EtO) is an important industrial compound, and over 11 million tons of EtO on an annual basis have been produced to manufacture plastics, textiles, and antifreeze [1]. EtO is also a new-generation surface disinfectant, extensively employed in the sterilization of medical equipment. On the other hand, EtO is an industrial chemical with high toxicity, posing a significant threat to both human health and the environment due to occasional or deliberate emissions [2]. Therefore, the development of advanced materials for protection against EtO exposure remains to be a momentous concern [3].

Personal protective equipment, such as respirator canisters, is indispensable for individuals to breathe safely in contaminated environments [4]. Activated carbons (AC) or impregnated activated carbons are frequently utilized as critical adsorbents for air purification owing to their high surface area and porosity, abundant pore regimes, excellent adsorption capacity, outstanding stability, environmental friendliness, large-scale availability, and adequate mechanical strength [5,6,7]. Although AC-based materials can provide protection against the toxic vapors of large molecules, they show limited efficacy in removing small molecular chemicals such as EtO due to their limited affinity of the adsorbate with the adsorbent, surface chemistry, and morphology [8,9,10]. 

Compared with AC, molecular sieves with nanoporous aluminosilicate framework structures have the advantages of structural stability, corrosion resistance, and high temperature resistance, which can be widely utilized in the fields of adsorption and catalysis [11,12,13]. Among them, ZSM-5 has a typical MFI topology and specific Si/Al ratios, exhibiting a unique cage structure and abundant Brønsted acid sites for enhancing the adsorption and catalysis of EtO [14]. However, ZSM-5 can be self-poisoned by the accumulation of reaction products in its pore owing to the low specific surface area and single pore structure [15].

In recent years, AC-based composites have been extensively studied due to the limitations of the pore structure and surface chemistry of various single adsorbent materials [9,16,17,18,19]. AC-based composites exhibit tremendous potential for application in various fields such as adsorption, catalytic degradation, electrochemical capacitors, and biosensors [20,21,22,23]. Zhang et al. prepared a series of Zeolite X/AC composites via the hydrothermal method, utilizing asphalt powder and silica as additional carbon and silica sources [24,25]. The HAX-3 composite, modified by ammonium chloride solution, displayed high selectivity for CH_4_/N_2_ (3.4) and an adsorption capacity for CH_4_ during pressure swing adsorption processes. Zhao et al. used a novel hybrid honeycomb monolith (H-ZSM-5/AC) as an adsorbent to capture CO_2_ through vacuum and electro swing adsorption (VESA) technology [26]. The hybrid material presented higher CO_2_ uptake and selectivity below 20 kPa pressure compared to commercial granular activated carbon. However, to our knowledge, there have been few systematic studies addressing the EtO adsorption on AC matrix composites.

In this work, a series of AC/ZSM-5 composites were facilely synthesized via mechanical mixing and the extrusion method with carboxymethyl cellulose as binder (Figure 1). The adsorption performance of AC/ZSM-5 composites for EtO were investigated using dynamic breakthrough experiments under different humidity conditions. The AC/ZSM-5 composites also demonstrated exceptional reusability after five adsorption-desorption cycles. Additionally, the interaction mechanism between EtO and the composite surface was confirmed by density functional theory (DFT) calculations.

## 2. Experimental Sections

### 2.1. Materials 

Commercial powdered activated carbon (coconut shell charcoal, screened 120~180 mesh, 99%, China) and commercial powdered ZSM-5 (Si/Al = 50%, screened 300~325 mesh, 99%, China) were purchased from Ningxia Guanghua Qisi Activated Carbon Co., Ltd. Carboxymethylcellulose (98%, Shizuishan, China) was obtained from Tianjin Komeo Chemical Reagent Co., Ltd. Ethylene oxide (25,000 ppm, Tianjin, China) was purchased from Beijing Haipu Gas Co., Ltd. (Beijing, China).

### 2.2. Preparation of AC/ZSM-5 Composites

First, 2.7 g of carboxymethyl cellulose (CMC) was added to 100 mL of deionized water (DI) to obtain a slurry by stirring. Then, 45 g of AC and 5 g of ZSM-5 were added into a mixer to obtain a uniform mixture, followed by spraying CMC slurry into the mixer and stirring for 5 min. Next, the obtained mixture was extruded into noodle-like samples and dried at 120 °C. Finally, the sample was physically pulverized and sieved to a particle size of 10 to 20 mesh, which was denoted as AC/ZSM-10%. Similarly, AC/ZSM-20% and AC/ZSM-30% were obtained by adjusting the dosage of ZSM-5. The preparation procedures of the pure AC or ZSM-5 samples were similar to that of AC/ZSM-10%, but a single component of AC or ZSM-5 powder was added. The final samples were labeled AC-R and ZSM-R, respectively.

### 2.3. Material Characterization

The morphological characteristics of each sample were observed via a scanning electron microscope (SEM, ZEISS Gemini-SEM 300, Germany). X-ray diffraction patterns (XRD, Rigaku SmartLab SE, Japan) were used to characterize the phases of each sample. The thermal gravimetric analysis data under air atmosphere were obtained on a thermogravimetric analyzer (TGA, Netzsch STA 449 F3, Germany). A Fourier transform infrared spectrometer (FTIR, Thermo Scientific Nicolet iS20, USA) was used to investigate the composition of the samples. To determine the specific surface area and pore size of the samples, the automatic specific surface area and pore structure analyzer (BET, Micromeritics ASAP 2460, USA) was employed to perform N_2_ adsorption-desorption tests at 77 K. The adsorption-desorption of water vapor at 298 K was studied using a water vapor isothermal adsorption-desorption apparatus (TPD, Micromeritics 3Flex, USA).

### 2.4. Calculation Models and Methods

In this study, VASP was used for the density functional theory (DFT) calculation. The Equation (1) was utilized to calculate the adsorption energy of EtO molecules on the surface of the substrate:(1)$$E_{ads}=E_{surface+EtO}-\left(E_{surface}+E_{EtO}\right)$$
where *E*_surface_ and *E*_EtO_ represent the energies of the substrate surface and the free EtO molecule, respectively, and *E*_surface+EtO_ is the total energy of the adsorbate-surface complex. The negative value of *E*_ads_ indicates that the process is an exothermic reaction [27], while the positive value indicates no adsorption. The strength of the interaction increased as the absolute value of *E*_ads_ increased.

## 3. Results and Discussion

### 3.1. Structural Analysis

SEM images of the prepared samples are presented in Figure 2. AC-R possessed abundant macropores with an average size of 9.2 μm (Figure 2a and Appendix A), indicating that the dosage of binder was appropriate and did not cause pore blockage. Figure 2e displays that ZSM-R has a uniform structure. Compared to AC-R, AC/ZSM-10% composite displayed a rough surface due to the introduction of ZSM-5, as shown in Figure 2b,f. With the increase in ZSM-5 content, AC/ZSM-20% composite exhibited a large number of disordered and interlaced columnar crystal units on the surface of the AC (Figure 2c,g). Surprisingly, ZSM-5 particles distributed on the surface of AC did not cause obvious blockage of carbon pores. With the further increase in ZSM-5 content, ZSM-5 particles densely dispersed on the AC surface, resulting in most of the carbon pores clogging (Figure 2d,h).

As shown in Figure 3a, the XRD patterns of AC/ZSM-5 composites matched considerably with the AC-R and ZSM-R, indicating the successful preparation of AC/ZSM-5 composites. As the amount of ZSM-5 increased, the characteristic peaks of the composites became sharper, indicating higher ZSM-5 loading in the composite. These findings were consistent with the SEM results.

TGA analyses were performed to evaluate the thermal stability of the samples, as shown in Figure 3b. ZSM-R had an unprecedented thermal stability, and its mass loss was only ~2% due to water evaporation between 25 °C and 1000 °C. The AC/ZSM-5 composites exhibited a comparable weight loss of ~7% below 200 °C. This was primarily due to the decomposition of oxygen-containing groups on the AC surface and water evaporation. In the range of 200~350 °C, the mass loss values of AC-R, AC/ZSM-10%, AC/ZSM-20%, and AC/ZSM-30% were 20.6%, 16.2%, 12.3%, and 10.8%, respectively, which was mainly due to the decomposition and volatilization of part of the binder. Importantly, the AC/ZSM-5 composites demonstrated higher thermal stability than AC-R owing to the introduction of ZSM-5.

Figure 3c shows the FT-IR spectra of AC-R, ZSM-R, and AC/ZSM-5 composites. AC-R has no obvious absorption peaks, which is attributed to the removal of functional groups in the structure under carbonization and activation conditions. The characteristic peaks of ZSM-5 at 1220 cm^−1^ and 1090 cm^−1^ belong to the framework vibration and the stretching vibration of Al-O-Si bonds, respectively. The infrared peaks at 550 cm^−1^ and 800 cm^−1^ were ascribed to the stretching vibration of Si-O bonds. With the increase in ZSM-5 content, the characteristic peaks of ZSM-5 in the composites gradually increased, which corresponded well with the SEM and XRD results.

Figure 3d presents the N_2_ adsorption-desorption isotherms of the samples. All samples except ZSM-5 showed typical type IV hysteresis loops, indicating the presence of micropores and mesopores. As summarized in Table 1, as the ZSM-5 amount increased, the textural properties of the samples decreased. The specific surface area (S_BET_) of AC/ZSM-5 composites decreased from 1271 m^2^ g^−1^ to 856 m^2^ g^−1^ with the increase in ZSM-5 amount, which was attributed to the low S_BET_ of ZSM-5 and pore clogging. The total pore volume (V_total_) also showed the same trend as the S_BET_. The V_tot_ of AC/ZSM-5 composites decreased from 0.568 cm^3^/g to 0.383 cm^3^/g. As depicted in Figure 3e, the NLDFT pore size distribution of the AC/ZSM-5 composites exhibited three major microporous sizes at 0.58 nm, 0.67 nm, and 0.85 nm, respectively, which coincided with that of AC-R and ZSM-5. Significantly, both AC/ZSM-20% and AC/ZSM-30% generated new micropores at 0.5 nm through inter-particle stacking between the AC and ZSM-5, which may promote the significant increase in EtO adsorption capacity. Figure 3f displays the BJH pore size distributions of AC/ZSM-5 composites, indicating the existence of mesopores at about 2–4.5 nm that originated from the AC. Moreover, we find that the mesoporous structure can be well-maintained with increasing ZSM content, which was consistent with the SEM observation.

### 3.2. EtO Adsorption Breakthrough Experiments

The schematic diagram of the adsorption device is shown in Appendix A. The dynamic adsorption behavior of EtO on AC/ZSM-5 composites was investigated through breakthrough experiments at 25 °C and 50% RH. As depicted in Figure 4, the breakthrough time and adsorption capacity of AC-R were 8.5 min and 32.5 mg/g, respectively. Compared with AC-R, the breakthrough time and adsorption capacity of ZSM-R were as high as 33 min and 52.3 mg/g. This is attributed to the numerous Brønsted acidic sites of ZSM-5, which provide greater protection against EtO. The breakthrough time of AC/ZSM-5 composites steadily increased to 52 min and then decreased to 25.2 min with the further increase in ZSM-5 amount. The adsorption capacity also showed the same trend as the breakthrough time. Among them, AC/ZSM-20% demonstrated remarkable breakthrough time and adsorption capacity, outperforming AC-R, ZSM-R, and numerous adsorbents under the same conditions (Figure 5).

These astonishing findings can be ascribed to the synergistic effect of AC and ZSM-5 structural features. In addition, the size of the newly formed micropores in AC/ZSM-20% composite was 0.5 nm, which was comparable to the kinetic diameter of EtO (0.37 nm) and was also conducive to EtO adsorption. However, when the proportion of ZSM-5 in the composite was 30%, the breakthrough time and adsorption capacity of AC/ZSM-30% decreased due to the blockage of carbon pores and the reduction in S_BET_.

To evaluate the effect of moisture amount on adsorption performance, the dynamic breakthrough experiments of EtO on AC/ZSM-20% were conducted under different humidity conditions (Figure 6a,b). Under dry conditions (RH = 0%), AC/ZSM-20% displayed an excellent breakthrough time and adsorption capacity with values of 97.5 min and 145.1 mg/g, respectively. As relative humidity increased, the breakthrough time and adsorption capacity steadily decreased due to the competitive adsorption of water molecules and EtO molecules, which is in accordance with previous reports [28]. Interestingly, the water absorption of AC/ZSM-20% at P/P_0_ was moderate, making it a promising candidate in practical scenarios (Figure 6d). Figure 6c shows five adsorption-desorption cycles over the AC/ZSM-20% composite at 25 °C. The EtO uptake of the AC/ZSM-20% maintained about 97% of the fresh sample after five cycles. This result manifested that the pore structure can be almost maintained after five cycles. Besides, the FT-IR, XRD, BET and DFT pore size distributions also confirm this exceptional reusability of AC/ZSM-20% (Appendix A). 

### 3.3. Adsorption Experiments and Model Fitting

To further investigate the adsorption behavior of EtO, three kinetic models, such as the pseudo-first-order, pseudo-second-order, and Boltzmann, are used to mathematically simulate the dynamic breakthrough curves of AC/ZSM-20% under different humidity conditions, as shown in Figure 7 and Appendix A. These semi-empirical models can effectively describe the mass transfer process of gases in porous media [29,30,31,32]. Figure 7a shows the fitted curves of samples at 50% RH. Compared with the pseudo-second-order model, the pseudo-first-order and Boltzmann models better described the adsorption process of EtO (Appendix A), indicating that the interaction between EtO and adsorbent is mainly physical adsorption rather than chemical adsorption. Notably, the Boltzmann model shows higher correlation coefficients (R^2^) than that of the pseudo-first-order for all the adsorbents, suggesting that this model accurately can express the dynamic adsorption process of EtO on composites. Similarly, the adsorption behavior of EtO under different humidity condition displays the same trend (Figure 7b). Among them, the Boltzmann model showed higher correlation coefficients (Appendix A). Moreover, the equilibrium adsorption capacity (*q_e_*) gradually decreased with the increase in relative humidity, which was consistent with the experimental data.

### 3.4. Molecular Modeling

To clarify the interaction mechanism between the composites and EtO molecules, DFT calculation was used to study the adsorption of EtO on composites. The adsorption distance reflects the optimal interaction distance between adsorbed molecules and the adsorbent surface and plays a key role in the stability of adsorption configuration [33]. The negative value of adsorption energy indicated that the adsorption process was exothermic, which conforms to the general characteristics of adsorption [34]. In the EtO-carbon surface system, the adsorption energy and adsorption distance of EtO on the AC surface was −15.36 kJ/mol and 3.119 Å, respectively (Figure 8a). Obviously, the adsorption energy in this system was in the range of physisorption, indicating the existence of only physical adsorption between EtO molecules and AC. However, in the EtO-ZSM-5 adsorption system, the absorption energy of a single EtO molecule adsorbed on the ZSM-5(010) surface was −51.84 kJ/mol due to the chemisorption of EtO molecules with ZSM-5 (Figure 8b). Compared with the above two systems, the adsorption energy of AC/ZSM-20% composite for EtO was particularly high with a value of −207.36 kJ/mol (Figure 8c), indicating that there was a strong interaction between EtO and AC/ZSM-20% composite due to their excellent physical and chemical comprehensive protective effects. Furthermore, the saturated adsorption capacity of the adsorbent increased with the increase in adsorption energy (Figure 8d), which corresponded well with the previous report [35].

## 4. Conclusions

A series of AC/ZSM-5 composites were produced by a facile mechanical mixing and extrusion method and investigated for EtO adsorption by dynamic breakthrough experiments. The AC/ZSM-20% exhibited the maximum EtO breakthrough time of 52 min and uptake amount of 81.9 mg/g at 25 °C and 50% RH, surpassing AC-R, ZSM-R, and representative adsorbents. In addition, the EtO adsorption capacity on the AC/ZSM-20% was maintained after five adsorption-desorption cycles, revealing its high reusability of up to ~97%. EtO molecules interacted with the composites via synergistic physisorption and chemisorption, as proved by DFT calculations and the Boltzmann kinetic model. This provided a solid foundation for the development of novel AC-based adsorbents as future toxic industrial chemicals adsorption media.

## Figures and Tables

**Figure 1 nanomaterials-14-01482-f001:**
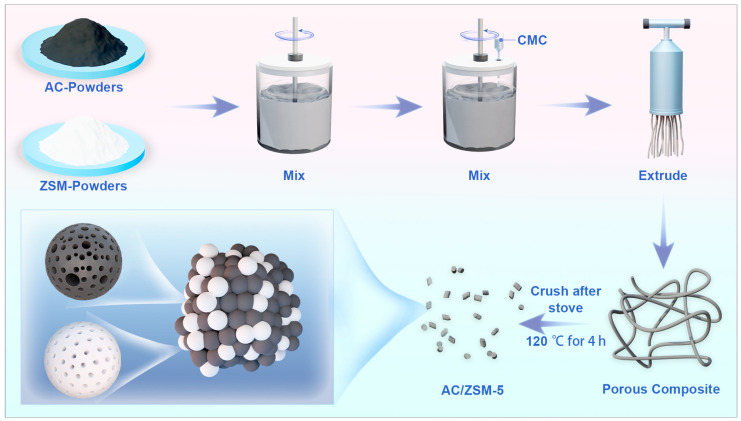
Schematic diagram of the synthesis of the AC/ZSM-5 composite.

**Figure 2 nanomaterials-14-01482-f002:**
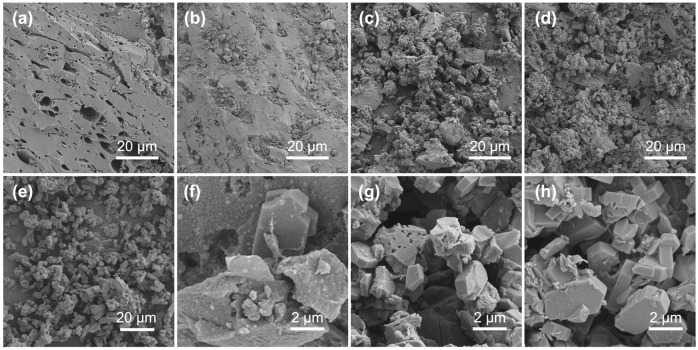
SEM images of (**a**) AC-R; (**e**) ZSM-R; (**b**,**f**) AC/ZSM-10% at different magnifications; (**c**,**g**) AC/ZSM-20% at different magnifications; and (**d**,**h**) AC/ZSM-30% at different magnifications.

**Figure 3 nanomaterials-14-01482-f003:**
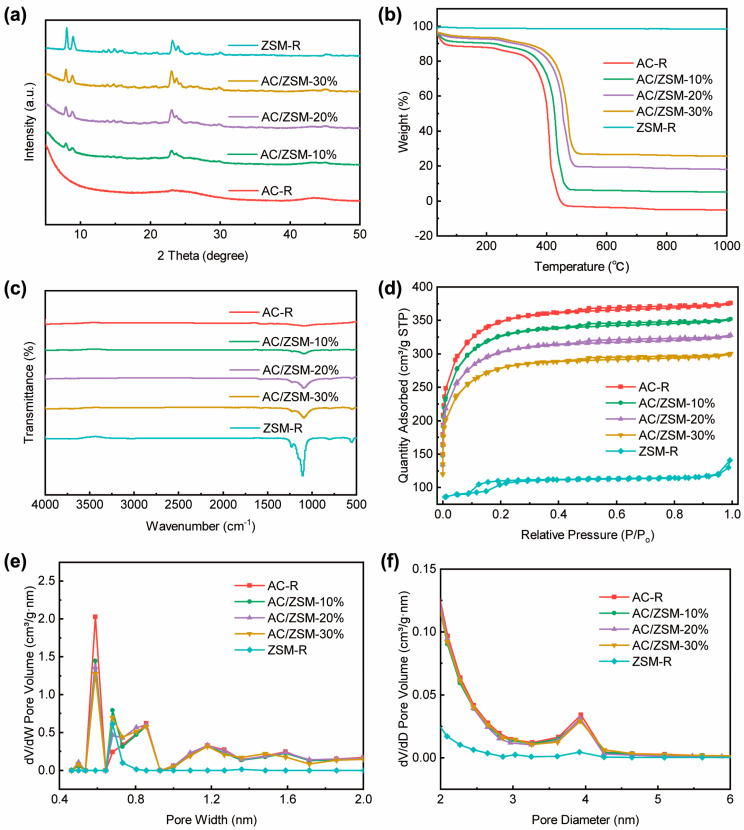
(**a**) XRD patterns, (**b**) TGA curves, (**c**) FT-IR spectra, (**d**) N_2_ adsorption-desorption isotherms, (**e**) DFT pore size distributions, and (**f**) BJH pore size distributions of samples.

**Figure 4 nanomaterials-14-01482-f004:**
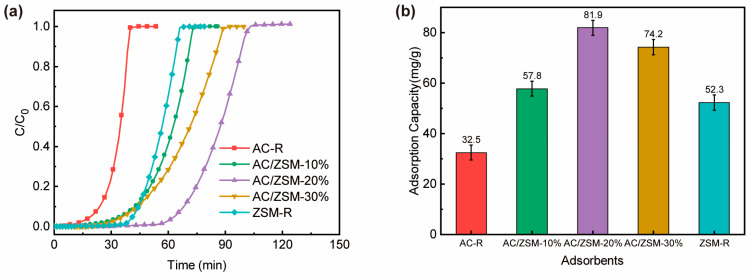
Comparison of dynamic adsorption performance of EtO at 25 °C and 50% RH (**a**) breakthrough curves and (**b**) histograms of calculated EtO adsorption capacity.

**Figure 5 nanomaterials-14-01482-f005:**
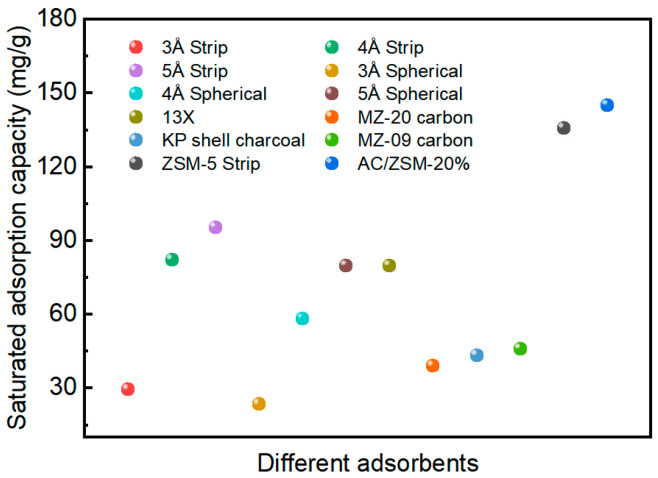
Comparison of saturated adsorption capacity of EtO adsorbed by different adsorbents; data are processed according to Appendix A.

**Figure 6 nanomaterials-14-01482-f006:**
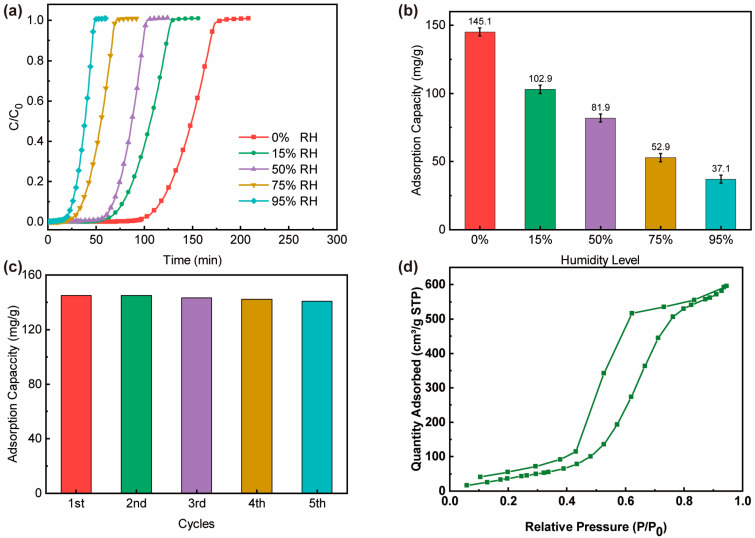
Comparison of dynamic adsorption performance of AC/ZSM-20% for EtO at different humidity levels (**a**) breakthrough curves, (**b**) EtO adsorption capacity, (**c**) calculated adsorption capacity for five cycles at 0% RH and (**d**) water vapor adsorption-desorption isotherm at 25 °C.

**Figure 7 nanomaterials-14-01482-f007:**
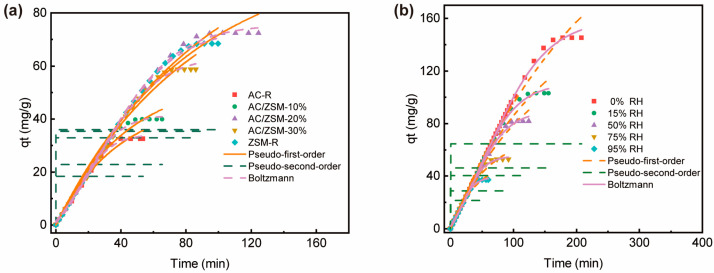
Mathematical adsorption simulation curves for (**a**) AC/ZSM-5 composites and (**b**) AC/ZSM-20% under different humidity conditions.

**Figure 8 nanomaterials-14-01482-f008:**
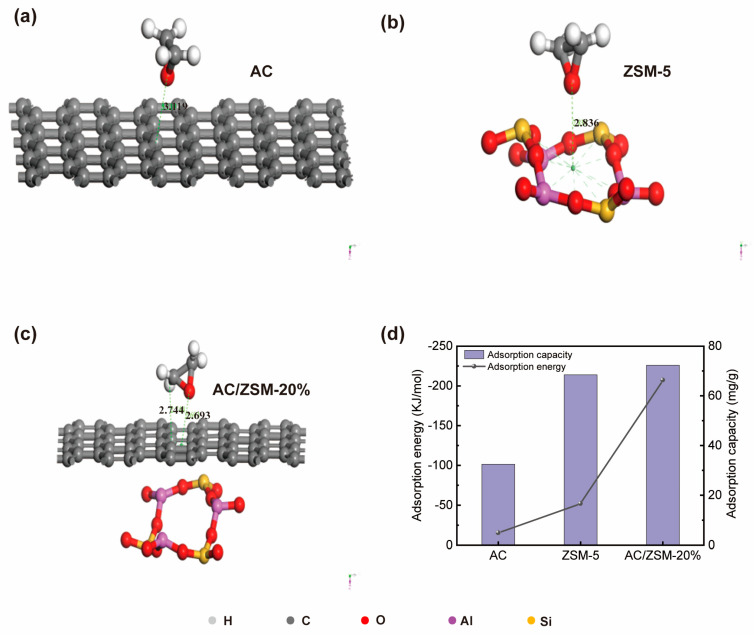
Comparison of DFT-calculated adsorption performance of AC, ZSM-5, and AC/ZSM-20% for EtO. (**a**) AC adsorption configuration, (**b**) ZSM-5 adsorption configuration, (**c**) AC/ZSM-20% adsorption configuration, and (**d**) comparison of adsorbent configuration and adsorption capacity.

**Table 1 nanomaterials-14-01482-t001:** Specific surface area (S_BET_), micropore surface area (S_mic_), total pore volume (V_tot_), and micropore volume (V_mic_) of samples.

Samples	S_BET_ (m^2^/g)	S_mic_ (m^2^/g)	V_total_ (cm^3^/g)	V_mic_ (cm^3^/g)
AC-R	1375.0	767.3	0.568	0.330
ZSM-R	364.9	123.0	0.184	0.053
AC/ZSM-10%	1004.6	603.6	0.453	0.324
AC/ZSM-20%	931.8	607.4	0.421	0.308
AC/ZSM-30%	856.1	524.8	0.383	0.281

## Data Availability

The data that has been used is confidential.

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
