# Peer review of "Introducing Molecular Sieve into Activated Carbon to Achieve High-Effective Adsorption for Ethylene Oxide"

_nanomaterials, 2024, doi:10.3390/nano14181482_

Round 1

Reviewer 1 Report

Comments and Suggestions for Authors

This manuscript deals with preparing activated carbon (AC) / ZSM-5 composites and their use in the ethylene oxide (EtO) adsorption from air. The effects of the AC/ZSM-5 ratio and the relative humidity of the air steam are studied. The materials were characterized by SEM, XRD and N2 adsorption-desorption and adsorption modelling has been applied. This is a good and interesting work which deserves to be published.

Some remarks:

1.    Section 2.2: It is not clear what AC-R and ZSM-R are. Are these the pristine materials? Do they also contain the binder?

2.    In the inner captions of Figure 1, we can read “pressurized mixing”, but the application of pressure is not reported in the text.

3.    Page 4, lines 150-154: In the text, it is said that Fig 3b shows the TGA curves of AC-R and ZSM-R. However, in the inner caption is just written AC and ZSM. Is it the same?

4.    Page 6, lines 201-202: I suggest that “inlet” and “outlet” should be used instead of “import” and “export”.

Comments on the Quality of English Language

The English usage is generally good. However, some mistakes need correction, such as:

Page 2, line 50: Instead of “…ZSM-5 molecular sieves will self-poisoned…”, it should be “…ZSM-5 molecular sieves will be self-poisoned…”

Page 2, lines 64-65: In the sentence “The novel hybrid integer showed higher CO2 adsorption and selectivity in the pressure range below 20kPa compared to commercial activated carbon integers” what do the authors mean by “integer”? This word is usually applied to numbers.

 Page 5, lines 164-165: In the sentence “the peak at around 1220 cm−1 corresponding to the framework vibration of ZSM-5 become strong”, it should be “becomes”.

Author Response

Dear reviewer, we have made revisions and responses based on your suggestions and feedback. 

Reviewer 2 Report

Comments and Suggestions for Authors

The paper deals with:

By introducing molecular sieve into activated carbon to achieve high-effective adsorption for ethylene oxide’,

Co-written by 9 authors:

Feng Liu, Lingyan Qin, Pingwei Ye, Bo Yang, Qiong Wu, Li Li, Yuwei Dai, Chuan Zhou and Sumin Li

This paper explores the effective ways to protect against the threat (human health and environment) from ethylene oxide (EtO).

Activated carbones (AC) may adsorb EtO, performance improved with ZSM-5 molecular sieve.

Homogeneous pores and surface Brønsted acidic sites of MS, coupled with the outstanding physisorption merit of AC itself, shows the adsorption performances of AC@ZSM-5 for EtO. Adsorption behaviors under different humidity conditions were investigated, and the adsorption properties were calculated according to the density-functional theory (DFT) model.

General comments: please add a glossary of abbreviations

TitleBy introducing molecular sieve into activated carbon to achieve high-effective adsorption for ethylene oxide’

What about this title?

Achieving highly efficient adsorption of ethylene oxide by introducing a molecular sieve into the activated carbon

1. Introduction

P2/16, Line 50. What about to modify the sentence ‘…ZSM-5 molecular sieves will self-poisoned by the accumulation of the reaction products within the pores of the zeolite blocks the pores [15].’

By

“…ZSM-5 molecular sieves will self-poison through the accumulation of reaction products in the pores of the pore-blocking zeolite [15].”

2. Experimental sections

2.1 Materials

P2/16, Line 81. In ‘Carboxymethylcellulose (AR) was purchased from…’ what is AR for?

2.2. Preparation of AC@ZSM-5 composites

P2/16, Line 84. Question: Why ‘carboxymethyl cellulose (CMC)’ is used?

P2/16, Line 91. Question: In ‘AC-Reborn’ and ‘ZSM-5-Reborn’, what is ‘Reborn’ for?

P3/16, Figure 1. During synthesis, first are mixed ‘powders of AC & ZSM’, then mixture is mixed with CMC [5%wt of (2.7g+100mL deionized water)].

About [5%wt of (2.7g+100mL deionized water)], 5% is for CMC weight or for the slurry?

About ‘pressurized mixing’, what ‘pressurized’ means?

3. Results and discussion

3.1. Structural analysis

P4/16, Line 123. Question: ‘…abundant macropores with a size of several microns to tens of microns…’, the authors can estimate the size and/or form profile of the macropores?

P4/16, Line 126. ‘…Fig2e…hexagonal shape…’ the reviewer does not really see the hexagonal shape and the reviewer does not understand why ‘hexagonal shape’ description is used.

Idem with ‘smooth surface’.

P5/16, Line 173. About ‘specific surface area’, can the authors describe the way to calculate it please (according to P3/16, Line 103. ‘porosity analyzer (Micromeritics ASAP 2460, USA)’).

P5/16, Line 178. About ‘Specific surface area (SBET), micropore surface area (Smic), total pore volume (Vtot), micropore volume (Vmic)’, can the authors recall the definitions for each, please.

P5/16, Line 181. About ‘three major microporous sizes at 0.58, 0.67 and 0.85 nm, respectively’ in Fig.3e, what is the error? What about 1.2 and 1.6? AC seems to not have 0.67family, isn’t it?

P5/16, Line 186 and Fig.3f. How dV/dP is calculated? What about the error estimations? BJH is for dV/dP?

3.2. EtO adsorption breakthrough experiments

P6/16, Paragraph Line 211. What about to add a Figure plotting equilibrium adsorption (on left axis) & family sizes pores (on right axis) versus materials? And notice when Brønsted acidic sites are activated.

Question: what about size family’s proportion competition in adsorption?  

P6/16, Line 247. The error estimation is calculated with two experiments?

P8/16, Line 268. ‘This is due to the strong 268 hydrogen bonding of water molecules, which competes with EtO molecules…’ Open question: Competition is one H2O/ one EtO for each one Brønsted acidic site/Brønsted acidic site? Or H2O will compete more because of hydrogen bonding? But hydrogen bonding is weaker or not compare to bonding with EtO?

3.3. Adsorption experiments and model fitting

P10/16, Lines 341. ‘These results indicate that high humidity environments are unfavorable for the capture of EtO.’ Ok, but Line 324 ‘the interaction between EtO molecules and composites is mainly based on physical adsorption rather than chemical adsorption.’

Open questions: What about the competition physical/chemical?

3.4. Molecular modeling

P11/16, Line 348. ‘DFT’ is?

Conclusion of the paragraph is? Can we suggest that competition physical/chemical is a variable confirmed by the adsorption energy calculation? A schematic competition physical/chemical rules versus adsorption energy could be possible?

References

Maybe need to be completed, as checked here after. Please, check also References in the “Supporting Information”  

1. DOI: 10.1021/acs.iecr.3c00402

2. DOI: 10.1002/ajim.23115

3. Environ. Technol. Innovation 15 (2019) 100376. DOI : 10.1016/j.eti.2019.100376

4. DOI: 10.1039/d3ta06108f

5. Appl. Surf. Sci. 508 (2020) 145211. DOI: 10.1016/j.apsusc.2019.145211

6. DOI: 0.1016/j.jclepro.2016.02.084

7. J. Hazard. Mater. 392 (2020) 122323. DOI: 10.1016/j.jhazmat.2020.122323

8. J. Nat. Gas Sci. Eng. 95 (2021) 104124. DOI: 10.1016/j.jngse.2021.104124

9. Water Res. 177 (2020) 115768. DOI: 10.1016/j.watres.2020.115768

10. J. Cleaner Prod. 302 (2021) 126925. DOI: 10.1016/j.jclepro.2021.126925

11. Chem. Eng. J. 390 (2020) 124513. DOI: 10.1016/j.cej.2020.124513

12. DOI: 10.1021/acs.energyfuels.8b02978

13. Appl. Therm. Eng. 213 (2022) 118746. DOI: 10.1016/j.applthermaleng.2022.118746

14. DOI: 10.1016/j.ces.2018.09.050

15. DOI: 10.1039/d0ra06040b

16. DOI: 10.1016/j.jiec.2022.08.021

17. DOI: 10.1016/j.biortech.2018.08.037

18. J. Cleaner Prod. 447 (2023) 138006. DOI: 10.1016/j.jclepro.2023.138006

19. J. Cleaner Prod. 325 (2021) 129271. DOI: 10.1016/j.jclepro.2021.129271

20. Int. J. Heat Mass Transfer 171 (2021) 121112. DOI: 10.1016/j.ijheatmasstransfer.2021.121112

21. Appl. Catal. B. 83(1-2) (2008) 63-71. DOI: 10.1016/j.apcatb.2008.02.003

22. Chem. Eng. J. 413 (2021) 127384. DOI: 10.1016/j.cej.2020.127384

23. Chem. Eng. J. 410 (2021) 128412. DOI: 10.1016/j.cej.2021.128412

24. DOI: 10.1016/j.jtice.2016.02.004

25. DOI: 10.1007/s10450-016-9836-3

26. DOI: 10.1016/j.cej.2018.09.196

27. Chem. Eng. J. 481 (2024) 148391. DOI: 10.1016/j.cej.2023.148391

28. Chem. Eng. J. 482 (2024) 148982. DOI: 10.1016/j.cej.2024.148982

29. Chem. Eng. J. 452 (2023) 139399. DOI: 10.1016/j.cej.2022.139399

30. Chem. Eng. J. 451 (2023) 138735. DOI: 10.1016/j.cej.2022.138735

31. Chem. Eng. J. 428 (2022) 131076. DOI: 10.1016/j.cej.2021.131076

32. Chem. Eng. J. 473 (2023) 1 44942. DOI: 10.1016/j.cej.2023.144942

33. Chem. Eng. J. 474 (2023) 145633. DOI: 10.1016/j.cej.2023.145633

34. DOI: 10.1016/j.cej.2019.04.218

Author Response

(The authors gave the same response as above.)

Reviewer 3 Report

Comments and Suggestions for Authors

The article approaches the introduction of molecular sieve into activated carbon to achieve high-effective adsorption for ethylene oxide.

The theme is interesting, although some concerns must be addressed:

- Please provide full information about the used reagents (purity/concentration, grade, company, city, country).

- All figures must be included after their first mention in the text.

- Please provide separate files for FTIR, TG and adsorption isotherms or regroup them in a more appropriate manner.

- Please replace “@” from “AC@ZSM-5” with a more appropriate symbol.

- Equations (2) and (3) must be edited with Math Editor provided by Microsoft Office Word.

- How the authors explain the reversed peak at about 3500 cm-1 from Fig S3.(a) – AC/ZSM-20%?

- Please provide the sources of the adsorbent samples from table S1 – the samples prepared within the study are not available within the tables, instead, other molecular sieves are mentioned, without providing their source. If these are from literature, please introduce the corresponding references or specify that they are commercial (company, city, country).  

- The verb time must be correlated: whether the Present or Past Tense is used, not a combination of them.

- The “Conclusion section” must be improved. 

Comments on the Quality of English Language

- The English language must be improved: English UK or US must be used, not a combination.

Author Response

(The authors gave the same response as above.)

Round 2

Reviewer 2 Report

Comments and Suggestions for Authors

Dear Authors. Thanks. Sincerely yours.

Reviewer 3 Report

Comments and Suggestions for Authors

The manuscript can be accepted in its present form.